# Effects of Initial Periodontal Therapy on Heat Shock Protein 70 Levels in Gingival Crevicular Fluid from Periodontitis Patients

**DOI:** 10.3390/jcm9103072

**Published:** 2020-09-24

**Authors:** Nobuhisa Furuse, Hideki Takai, Yorimasa Ogata

**Affiliations:** 1Department of Periodontology, Nihon University School of Dentistry at Matsudo, Matsudo, Chiba 271-8587, Japan; furuse-dental@jcom.zaq.ne.jp (N.F.); takai.hideki@nihon-u.ac.jp (H.T.); 2Research Institute of Oral Science, Nihon University School of Dentistry at Matsudo, Matsudo, Chiba 271-8587, Japan

**Keywords:** 70-kDa heat shock proteins, gingival crevicular fluid, heat shock protein, periodontitis

## Abstract

Periodontitis is an inflammatory disease of periodontium which is caused by periodontopathic bacteria. Moreover, various cytokines such as interleukin-1β (IL-1β), tumor necrosis factor-α (TNF-α), and IL-6 are expressed in the inflamed periodontium. Heat shock proteins (HSPs) protect cells from abnormal conditions including inflammation, microbial infection and diseases. The 70-kDa HSPs (HSP70s) are major HSPs that express in the inflamed tissues. In this study, an enzyme-linked immunosorbent assay was applied to measure the levels of HSP70 in gingival crevicular fluid (GCF) from two periodontal pockets in each of 10 patients with Stage III, Grade B periodontitis. Sites with probing pocket depth (PPD) of ≤3 mm were named the healthy control (HC) sites, and sites with PPD of ≥5 mm were named the diseased sites. HSP70 levels in GCF were expressed higher at diseased sites than at HC sites, and decreased after initial periodontal therapy at diseased sites. These results suggest the association of HSP70 with the stage of periodontitis.

## 1. Introduction

Periodontitis, a common disease, is the inflammation of periodontium caused by oral microorganisms [1]. Numerous cytokines have been detected in the gingival crevicular fluid (GCF). Interleukin-1β (IL-1β) and tumor necrosis-α (TNF-α) levels have been found in increased GCF from inflamed gingival tissues [2,3]. IL-1β, matrix metalloproteinase-8, and IL-8 levels in GCF were significantly higher in diseased sites than in healthy subjects, and those levels in GCF were significantly decreased after periodontal treatment [4]. In a recent study, GCF volume and IL-1β levels in GCF reflected the disease severity, and these parameters were suggested to be better than probing pocket depth (PPD) and bleeding on probing (BOP) as markers of gingival inflammation [5]. Therefore, it could be helpful to investigate the inflammatory cytokine levels in GCF for diagnosis of the active phase of periodontal disease.

Heat shock proteins (HSPs) act as molecular chaperones, which are enhanced proteins for protecting cells immediately after cells are exposed to heat shock stress [6]. The 70-kDa HSPs (HSP70s) comprising various isoforms are involved in a number of human pathologies, ranging from cancer to neurodegenerative diseases [7,8]. HSP70 levels were higher in protein extract from inflamed gingiva collected during periodontal surgery, and HSP60 and HSP65 levels were higher in serum from the patients with periodontitis [9]. In addition, expression levels of IL-1β, IL-8, and HSP70 were increased in GCF from patients with periodontitis compared to healthy subjects [10]. Serum concentration of chitinase-3-like-1 (YKL-40), which is a novel marker of acute and chronic inflammation, was significantly higher in patients with periodontitis and diabetes compared to healthy groups. However, GCF concentration of YKL-40 was similar in patients with periodontitis, patients with diabetes, and healthy subjects. YKL-40 levels were significantly increased in diabetes patients with periodontitis [11]. Therefore, HSP70 can be considered potentially as a marker for the severity of periodontal disease. However, there is no report on changes in HSP70 concentration in GCF before and after initial periodontal therapy, or its correlation with clinical parameters. The aim of this study was to elucidate the HSP70 levels in GCF from patients with periodontitis, and to compare their concentrations in GCF at first visit, after initial periodontal therapy, and at a three month follow-up.

## 2. Materials and Methods

### 2.1. Study Population

Ten patients with Stage III, Grade B periodontitis (mean age, 53.9 ± 3 years) at Nihon University Hospital School of Dentistry at Matsudo, Japan were recruited and then received initial periodontal therapy, including oral hygiene instruction, scaling and root planing, and professional mechanical tooth cleaning. The stage of periodontitis was assessed clinically based on clinical attachment loss (CAL), radiographic bone loss (BL), periodontal history of tooth loss (PTL), PPD, BOP, furcation involvement (Class II or III), plaque index (PlI), and gingival index (GI) [12,13,14]. A CP11 probe (Hu-Friedy, Chicago, IL, USA) was used to measure the PPD and CAL. They were diagnosed when BL affected the middle third of the root or beyond and CAL was 5 mm or more. If PTL was four teeth or less, then the diagnosis was stage III. When the patient’s previous periodontal records were available, the rate of periodontitis progression in the previous five years could be estimated. If progression was <2 mm, then the diagnosis was Grade B periodontitis [15]. GCF samples were taken from two periodontal PPD sites (shallow PPD of ≤3 mm were named the healthy control (HC) sites and deep PPD of ≥5 mm were named the diseased sites) for each patient.

This study was approved by the ethics committee at Nihon University School of Dentistry at Matsudo (EC17-017; 6Dec2017, EC18-17-017-1; 31, August, 2018). Written informed consent was obtained from all patients after all of the details of the investigative approach had been explained to them. Ten patients were confirmed physically healthy and had no experience of periodontal treatment or antibiotic treatment for at least three months prior to participation in this study. None of the patients were smokers.

### 2.2. Sample Size

Sample size was determined by the statistical software Easy R (EZR) on R commander Ver.1.27 (Tokyo, Japan) [16]. It was calculated that six samples (samples were taken from two sites per patient) were necessary for each group (HC and diseased sites) to reach 80% power at 5% level of significance. Therefore, the number of samples taken was appropriate.

### 2.3. Clinical Protocol

Clinical examinations were performed three times (first visit (1st), second examination (2nd) after initial periodontal therapy, and third examination (3rd) at a three month follow-up after initial periodontal therapy) by two periodontal specialists (H.T. and Y.O.). Average period of initial periodontal therapy was three months. At the point of each examination (1st, 2nd, and 3rd), GCF was collected using Periopaper subsequent to removal of supragingival plaque with a sterile curette. The GCF volume (μL) was measured using a Periotron 4000 (Oraflow, New York, NY, USA). GCF samples for the second visit data were taken before the periodontal therapy on the same day. Periopaper was then inserted into the pocket for 30 s each time. GCF samples were kept in Eppendorf tubes from two periodontal sites (≥5 mm (deep diseased site) and ≤3 mm (shallow healthy control site; HC)) from each patient and stored at −80 °C until measurement of HSP70 levels [17].

### 2.4. Enzyme-Linked Immunosorbent Assay (ELISA)

Concentrations of HSP70 in GCF samples were measured by ELISA using the Human HSPA4 (HSP70) ELISA kit (Invitrogen) according to the manufacturer’s instructions. GCF samples were dissolved in 300 μL of Sample Diluent C in the ELISA kit. Diluted samples (100 μL) were incubated for 2.5 h in an anti-human HSP70 pre-coated 96-well strip plate. After a wash, biotinylated antibody was added to the wells for 1 h. After a wash, streptavidin-HRP solution was added to the wells for 45 min. After washing, TMB substrate was added to the wells for 30 min. Color development was stopped and the optical density at 450 nm of each well was measured within 30 min using a microplate reader. All measurements were performed in duplicate and the concentrations of HSP70 were expressed in ng/mL.

### 2.5. Statistical Analysis

Clinical parameters are shown as mean ± standard error (SE) in Table 1. The significant differences between each examination (1st, 2nd, and 3rd) for clinical parameters, GCF volume, and HSP70 levels among the groups were determined by one-way ANOVA. Difference in BOP at the 1st to 3rd examinations were analyzed by a Chi-squared test. The level of significance was adjusted at 5%. Four steps Ekuseru-Toukei (the publisher OMS Ltd.) was used for statistical analyses.

## 3. Results

The patient characteristics such as age, sex, PPD, CAL, PlI, GI, and BOP distributions for the 10 patients in this study are listed in Table 1. Average PPD and CAL at the HC sites (PPD ≤ 3 mm) were 2.7 ± 0.2 mm and 3.9 ± 0.4 mm, and at the diseased sites (PPD ≥ 5 mm) were 6.5 ± 0.5 mm and 7.7 ± 0.7 mm, respectively. GI and BOP scores at the diseased sites (1.7 ± 0.2 and 80%) were higher than those at the HC sites (0.3 ± 0.2 and 0%). PlI at the HC and diseased sites were the same score (1.1 ± 0.2).

The concentrations of HSP70 in GCF from the HC and diseased sites at each point of examination during initial periodontal therapy are shown in Table 2. The average HSP70 level at the 1st visit in GCF from diseased sites was significantly higher than HC sites. Moreover, the concentration of HSP70 at diseased sites was significantly decreased at 3rd examination (a three month follow-up after initial periodontal therapy) as compared to the 1st examination. The concentration of HSP70 at HC sites did not change through the periodontal therapy (1st, 2nd, and 3rd examinations) (Table 2).

Changes in five kinds of clinical parameters (PPD, CAL, PlI, GI, and BOP scores) at the HC and diseased sites during initial periodontal therapy are listed in Table 3 and Table 4. At HC sites, PlIs were significantly decreased at the 2nd and 3rd examinations as compared to the 1st examination (Table 3). On the other hand, PPD and PlI at diseased sites were significantly decreased at the 2nd and 3rd examinations compared to the 1st examination. Furthermore, GI and BOP scores at diseased sites were significantly decreased at the 3rd examination compared to the 1st examination. GCF volumes from HC and diseased sites were measured by Periotron 4000 during the course of periodontal therapy (Table 5). GCF volumes from HC and diseased sites did not change during the periodontal therapy, however, the volumes of GCF from the diseased sites were significantly higher than those in the HC sites at the 1st and 2nd visits (Table 5).

## 4. Discussion

In this study, we have shown that there was a significant difference in HSP70 concentration in GCF between HC and diseased sites at the first visit. The concentration of HSP70 in GCF from diseased sites was significantly decreased at the three month follow-up after initial periodontal therapy (3rd examination; Table 2). At HC sites, PlI was significantly decreased at the 2nd and 3rd examinations (Table 3). At diseased sites, PPD and PlI were significantly decreased at the 2nd and 3rd examinations, whereas GI and BOP were significantly decreased only at the 3rd examination as compared to the 1st visit (Table 4). These results suggest that initial periodontal therapy is effective in improving inflammation of periodontal tissues and there is an association between the level of HSP70 and periodontitis. In addition, improvements of GI, BOP, and HSP70 levels were found to take longer than improvements of PPD and PlI.

Intracellular HSP levels are elevated immediately after exposed to stresses such as high temperature. HSPs are involved in the maintenance of cellular homeostasis and protein repair in damaged cells [18]. However, there are several unclear points in the relationship between HSPs and periodontitis. Inflammatory periodontal pockets have a higher temperature than healthy pockets [19]. Inflammatory cytokines, such as IL-1, TNF-α, and INF-γ, are produced in inflamed periodontal tissues [20], and they might act as stressors to induce the expression of HSPs. Lipopolysaccharide (LPS) and IL-1 increased hyperthermia-induced HSP70 in monocyte/macrophage-like RAW264.7 cells [21]. However, one study described how HSP70 dramatically down-regulated in the inflamed periodontal tissues [22]. Another study showed that GCF volume at the first visit decreased significantly after initial periodontal therapy [23]. However, in this study, GCF volumes from HC and diseased sites did not change during periodontal therapy (Table 5), although the GCF volumes from diseased sites at the 1st and 2nd visits were significantly higher than the GCF volumes from HC sites (Table 5). Therefore, further study is necessary to elucidate the involvement of HSP70 in the onset and progression of periodontitis.

Stress and smoking are environmental factors for periodontitis [24,25]. Several studies have shown that smoking has an adverse effect on the incidence and progression of periodontitis [25]. In the synovial tissues of smokers with rheumatoid arthritis (RA), HSP70 levels were significantly higher than in the synovial tissues of non-smokers with RA [26]. Therefore, smoking could increase the expression of HSP70. There are several studies describing the association between HSP70 and cancer. Malignant cells, such as osteosarcoma derived cells, expressed higher levels of HSP70 during tumor progression compared to normal cells [27]. Moreover, HSP70 has been assessed as a marker for oral epithelial dysplasia such as oral leukoplakia [28]. Therefore, various studies have been conducted to develop the HSP70 inhibitors for cancer therapy [29]. We have previously shown that the anti-HSP70 antibody levels were significantly higher in GCF from HC sites than diseased sites, and the anti-HSP70 antibody levels were increased after initial periodontal therapy in both HC and diseased sites [30]. Therefore, these results suggest that anti-HSP70 antibody may reduce inflammation of periodontal tissues via decreasing the HSP70 levels.

In conclusion, GCF volumes from the diseased sites were significantly higher than those in the HC sites at the 1st and 2nd visits. HSP70 concentration in GCF from diseased sites was significantly higher than the concentration of HSP70 from HC sites at the 1st visit. Moreover, the HSP70 concentration at the 1st visit was significantly decreased at the three month follow-up after initial periodontal therapy together with clinical parameters, such as PPD, GI, PlI, and BOP. These results suggest that the HSP70 concentration could become an appropriate indicator for the healing process of periodontitis.

## Figures and Tables

**Table 1 jcm-09-03072-t001:** Patient characteristics.

	HC Sites	Diseased Sites
Age	53.9 ± 3	
Gender (male/female)	2/8	
PPD	2.7 ± 0.2	6.5 ± 0.5
CAL	3.9 ± 0.4	7.7 ± 0.7
PlI	1.1 ± 0.2	1.1 ± 0.2
GI	0.3 ± 0.2	1.7 ± 0.2
BOP	0 (0%)	8 (80%)

HC, healthy control; PPD, probing pocket depth; CAL, clinical attachment loss; PlI, plaque index; GI, gingival index; BOP, bleeding on probing; mean ± SE (*n* = 10).

**Table 2 jcm-09-03072-t002:** Changes in the concentrations of HSP70 in GCF collected from HC and diseased sites during the periodontal therapy.

Concentration (ng/mL)	1st	2nd	3rd
HC sites	18 ± 4.99 *	16.73 ± 6.37	6.64 ± 3.46
Diseased sites	64.36 ± 13.74	35.1 ± 6.67	5.69 ± 1.78 **

GCF, gingival crevicular fluid; HC, healthy control; 1st, first visit; 2nd, after initial periodontal therapy; 3rd, a three month follow-up after initial periodontal therapy; mean ± SE (*n* = 10), * *p* < 0.05, ** *p* < 0.01.

**Table 3 jcm-09-03072-t003:** Changes in clinical parameters at HC sites.

	1st	2nd	3rd
PPD	2.7 ± 0.2	2.5 ± 0.2	2.5 ± 0.2
CAL	3.9 ± 0.4	3.6 ± 0.5	3.6 ± 0.4
PlI	1.1 ± 0.2	0.6 ± 0.2 *	0.6 ± 0.2 *
GI	0.3 ± 0.2	0	0.4 ± 0.3
BOP	0 (0%)	0 (0%)	2 (20%)

HC, healthy control; 1st, first visit; 2nd, after initial periodontal therapy; 3rd, a three month follow-up after initial periodontal therapy; PPD, probing pocket depth; CAL, clinical attachment loss; PlI, plaque index; GI, gingival index; BOP, bleeding on probing; mean ± SE (*n* = 10), * *p* < 0.05.

**Table 4 jcm-09-03072-t004:** Changes in clinical parameters at diseased sites.

	1st	2nd	3rd
PPD	6.5 ± 0.5	4.3 ± 0.4 *	4.3 ± 0.5 *
CAL	7.7 ± 0.7	6.2 ± 0.9	5.9 ± 0.7
PlI	1.1 ± 0.2	0.5 ± 0.2 *	0.6 ± 0.2 *
GI	1.7 ± 0.2	1.2 ± 0.3	0.9 ± 0.3 *
BOP	8 (80%)	5 (50%)	4 (40%) *

1st, first visit; 2nd, after initial periodontal therapy; 3rd, a three month follow-up after initial periodontal therapy; PPD, probing pocket depth; CAL, clinical attachment loss; PlI, plaque index; GI, gingival index; BOP, bleeding on probing; mean ± SE (*n* = 10), * *p* < 0.05.

**Table 5 jcm-09-03072-t005:** Changes in GCF volume at HC and diseased sites during the periodontal therapy.

GCF (μL)	1st	2nd	3rd
HC sites	0.92 ± 0.2	0.47 ± 0.2	0.66 ± 0.2
Diseased sites	2.41 ± 0.5 *	1.6 ± 0.3 *	1.31 ± 0.3

GCF, gingival crevicular fluid; HC, healthy control; 1st, first visit; 2nd, after initial periodontal therapy; 3rd, a three month follow-up after initial periodontal therapy; mean ± SE (*n* = 10), * *p* < 0.05.

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
