# Peer review of "Effects of Initial Periodontal Therapy on Heat Shock Protein 70 Levels in Gingival Crevicular Fluid from Periodontitis Patients"

_jcm, 2020, doi:10.3390/jcm9103072_

Round 1
Reviewer 1 Report
1) Chronic Periodontitis is no longer used in the new periodontal classification. Diseased sites suggested to be used.
2) Intial therapy needs to be defined
3) its not clear whether test samples are patients or sites.
4) its not clear how many sites/samples are in each group.
5) some claims in the discussion need to be referenced more accurately.
6) Please see the attached file for detailed edits.

Author Response
Reviewer 1
Dear Editor in Chief
Journal of Clinical Medicine
On behalf of all authors, I would like to thank you and reviewers for reviewing our manuscript. All the comments made by the reviewers were truly constructive and contributed to the further improvement of our revised manuscript.
Our responses to their comments are listed below. We extensively revised our manuscript accordingly and highlighted the changes within the document by using red colored text.
We believe that the changes that we have made in response to the constructive comments of the reviewers, have improved the presentation of this study, which we hope will now be considered acceptable for publication in Journal of Clinical Medicine.
Best regards,
Yorimasa Ogata, DDS, PhD
Department of Periodontology
Nihon University School of Dentistry at Matsudo
- Chronic Periodontitis is no longer used in the new periodontal classification. Diseased sites suggested to be used.
(Response)
According to the reviewer’s comment, we have changed the diagnosis name from chronic Periodontitis to Stage â…¢, Grade B periodontitis, and used diseased sites instead of CP sites. We have revised reference No. 14 and added new reference No.15.
- Initial therapy needs to be defined
(Response)
We have done the initial periodontal therapy including oral hygiene instruction, scaling and root planing, and professional mechanical tooth cleaning. We have added the description “and then received initial periodontal therapy including oral hygiene instruction, scaling and root planning, and professional mechanical tooth cleaning.” in page 2, lines 12~14.
- Its not clear whether test samples are patients or sites.
- Its not clear how many sites/samples are in each group.
(Response)
GCF samples were taken from two periodontal PPD sites (shallow PPD of ≤3 mm named the healthy control (HC) sites and deep PPD of >6 mm named the diseased sites) in each of 10 patients.
- Some claims in the discussion need to be referenced more accurately.
(Response)
According to the reviewer’s comment, we have added 2 references (No. 24 and 30).
- Please see the attached file for detailed edits.
(Response)
Thank you very much for your detailed edits. We revised according to your corrections.

Reviewer 2 Report
Overall, good project and decent writing. But need improvement and addition/correction in relation to the following aspects of the paper.
- Abstract/Introduction: Why the old Periodontal disease classification that is chronic periodontitis was used in this study instead of new 2018 classification.
- Material and Methods: Its nowhere mentioned what kind of initial therapy was performed. Was scaling and root planing done for CP sites and prophylaxis for HC sites etc.. please elaborate on this. Also mention in the study population how the HC sites were defined. Were there any smokers enrolled in the study.
- Clinical Protocol: Was second visit data collection done on the same day as the periodontal therapy visit? Please elaborate on this.
- Results: How would you explain the HSP70 change in concentration over period of 3 months in the HC sites as shown in table 2?
- What are the limitations of this study if any, do mention those in the discussion.
Author Response
Reviewer 2
Dear Editor in Chief
Journal of Clinical Medicine
On behalf of all authors, I would like to thank you and reviewers for reviewing our manuscript. All the comments made by the reviewers were truly constructive and contributed to the further improvement of our revised manuscript.
Our responses to their comments are listed below. We extensively revised our manuscript accordingly and highlighted the changes within the document by using red colored text.
We believe that the changes that we have made in response to the constructive comments of the reviewers, have improved the presentation of this study, which we hope will now be considered acceptable for publication in Journal of Clinical Medicine.
Best regards,
Yorimasa Ogata, DDS, PhD
Department of Periodontology
Nihon University School of Dentistry at Matsudo
Overall, good project and decent writing. But need improvement and addition/correction in relation to the following aspects of the paper.
(Response)
Thank you very much for your positive comment.
Abstract/Introduction: Why the old Periodontal disease classification that is chronic periodontitis was used in this study instead of new 2018 classification.
(Response)
According to the reviewer’s comment, we have changed the diagnosis name from chronic Periodontitis to Stage â…¢, Grade B periodontitis, and used diseased sites instead of CP sites. We have revised reference No. 14 and added new reference No.15.
Material and Methods: Its nowhere mentioned what kind of initial therapy was performed. Was scaling and root planing done for CP sites and prophylaxis for HC sites etc. please elaborate on this.
(Response)
Thank you very much for your comment. We have done the initial periodontal therapy including oral hygiene instruction, scaling and root planing, and professional mechanical tooth cleaning. We have added the description “and then received initial periodontal therapy including oral hygiene instruction, scaling and root planning, and professional mechanical tooth cleaning.” in page 2, lines 12~14. GCF samples were taken from two periodontal PPD sites (shallow PPD of ≤3 mm named the HC sites and deep PPD of >6 mm named the diseased (CP) sites) for each patient.
Also mention in the study population how the HC sites were defined. Were there any smokers enrolled in the study.
(Response)
GCF samples were taken from two periodontal PPD sites (shallow PPD of ≤3 mm named the HC sites and deep PPD of >6 mm named the diseased (CP) sites) for each patient. There is no smoker in the patients.
Clinical Protocol: Was second visit data collection done on the same day as the periodontal therapy visit? Please elaborate on this.
(Response)
GCF samples for the second visit data were taken before the periodontal therapy on the same day. We have added the description in page 2, lines 43~44.
Results: How would you explain the HSP70 change in concentration over period of 3 months in the HC sites as shown in table 2?
(Response)
The concentration of HSP70 at HC sites did not change through the periodontal therapy (1st, 2nd and 3rd examinations) (Table 2). There is a description in page 3.
What are the limitations of this study if any, do mention those in the discussion.
(Response)
According to the reviewer’s comment, we have added the description of the limitations of this study in the discussion.
